# Effects of Different Cooking Methods on the Vitamin D Content of Commonly Consumed Fish in Thailand

**DOI:** 10.3390/foods11060819

**Published:** 2022-03-12

**Authors:** Piyanut Sridonpai, Kunchit Judprasong, Nichaphan Tirakomonpong, Preecha Saetang, Prapasri Puwastien, Nipa Rojroongwasinkul, Boonsong Ongphiphadhanakul

**Affiliations:** 1Institute of Nutrition, Mahidol University, Salaya, Phutthamonthon, Nakhon Pathom 73170, Thailand; piyanut.sri@mahidol.ac.th (P.S.); pentail@hotmail.com (N.T.); white-phalaenopsis@hotmail.com (P.S.); prapasri.puw@mahidol.ac.th (P.P.); nipa.roj@mahidol.ac.th (N.R.); 2Faculty of Medicine, Ramathibodi Hospital, Mahidol University Ratchatewi, Bangkok 10400, Thailand; boonsong.ong@mahidol.ac.th

**Keywords:** vitamin D, freshwater fish, marine fish, cooking method, true retention

## Abstract

This study determined vitamin D content in commonly consumed fish in Thailand and the effects of different cooking methods on vitamin D retention. Five species of freshwater fish and four species of marine fish were purchased from three representative markets. All of the fish were individually prepared according to common household practices. Vitamin D2 and D3 were determined using the HPLC standard method (AOAC method 995.05). The results indicated that vitamin D3 was the only detectable form of vitamin D in the fish. Vitamin D content of raw freshwater fish ranged from 2.42 to 48.5 µg per 100 g edible portion (EP), which was higher than that of raw marine fish (2.94 to 4.69 µg per 100 g EP). Common silver barb, Red Nile tilapia, and Nile tilapia (freshwater fish living in the limnetic zone) contained high levels of vitamin D (48.5 ± 26.5, 31.0 ± 7.7, and 19.8 ± 3.5 µg per 100 g EP, respectively). Boiled fish (except for Common silver barb), fried fish (except for Striped snakehead, Walking catfish, and Common silver barb), and grilled fish (except for Common silver barb, Giant sea perch, and Short-bodied mackerel) retained high levels of vitamin D, which were not significantly different (*p* > 0.05) from raw fish. Common silver barb, Red Nile tilapia, and Nile tilapia—cooked by boiling, frying, and grilling—are recommended for consumption as excellent sources of vitamin D.

## 1. Introduction

Vitamin D is a prohormone produced in the skin through ultraviolet irradiation of 7-dehydrocholesterol that is metabolized to 25-hydroxyvitamin D3 in the liver and then to 1α,25-dihydroxyvitamin D3 in the kidney before function [1]. This hormone plays an important role in many bodily functions, including calcium absorption, calcium mobilization in bone, and calcium reabsorption in the kidney, and may well prevent several degenerative diseases [2]. 

The level of 25-hydroxyvitamin D3 [25(OH)D3] in serum is used to indicate vitamin D status among patients. In Thailand, a high prevalence of vitamin D insufficiency exists in several population groups, including children, adolescents, adults, and elders. Reesukumal et al. (2015) [3] studied the prevalence of hypovitaminosis D in 159 healthy children (aged 6 to 12 years) living in Bangkok, Thailand. They reported that the prevalence of vitamin D insufficiency (serum <75 nmol/L of 25(OH)D3) and vitamin D deficiency (serum < 50 nmol/L of 25(OH)D3) were 59.7% and 19.5%, respectively. For vitamin D insufficiency in Thai adults (2641 adults, aged 15–98 years), the prevalence according to geographical region was 43.1%, 39.1%, 34.2%, and 43.8% in the country’s central, northern, northeastern, and southern regions, respectively [4,5]. Healthy Thai females (aged 25–54 years) showed a three-fold higher prevalence of vitamin D deficiency than males (43.1% compared to 13.9%) [4]. Several factors may be causally related to lower vitamin D status, including lifestyle and environmental conditions, such as having indoor jobs, inadequate outdoor sun exposure, usage of sunscreen, and sun avoidant behavior. Food is an alternative source for increasing the level of vitamin D in humans. The most common dietary forms of vitamin D are vitamins D2 and D3. The D3 form is found in animal-sourced foods, such as fish oil and egg yolk [6]. In contrast, vitamin D2 is found mainly in yeast and plants [6]. 

The Thai Dietary Reference Intake (Thai DRI) of vitamin D for persons aged 1–70 years is 15 µg/day (600 International Unit, IU) [7]. The USDA databases show that fish, egg, and dairy products are the main naturally occurring food sources of vitamin D3, while mushrooms (fungi) are the main food source of vitamin D2 [6]. Several fish (e.g., halibut, mackerel, carp, eel, salmon) are the best sources of vitamin D3 (27.4, 25.2, 24.7, 23.3, and 21.5 μg per 100 g edible portion [EP], respectively) [6]. The vitamin D3 content of different fish species shows high variation due to several factors, such as environment, season, climate, age, food supply, and type of species [8]. Moreover, cooking methods may play an important role in determining the final nutrient contents in fish [9]. 

In Thailand, fish is the second most commonly consumed food. Fish has good quality protein and contains unsaturated fat, omega-3 unsaturated fatty acids, vitamins, and minerals. Tirakomonpong et al. (2019) [10] reported vitamin D content in ten species of freshwater (5 species) and marine (5 species) fish that are commonly consumed in Thailand. Their results showed that three types of raw freshwater fish, namely Common silver barb, Red Nile tilapia, and Nile tilapia, contained high amounts of vitamin D (48.5, 31.0, and 19.8 μg per 100 g EP, respectively), whereas raw marine fish provided low amounts of vitamin D (2.9 to 4.7 μg per 100 g EP). However, food composition databases on the vitamin D content of cooked fish and the effect of cooking on vitamin D are not available. Consequently, this study assessed vitamin D content in commonly consumed freshwater and marine fish in raw and cooked forms (boiling, frying, and grilling) and examined the true retention of vitamin D after different cooking methods. This information can be used to develop a vitamin D database for promoting the consumption of fish to prevent vitamin D deficiency in Thailand.

## 2. Materials and Methods 

### 2.1. Chemicals and Reagents

All reagents used were of analytical grade, whereas the organic solvents used for HPLC were of HPLC grade. Standards for vitamin D2 (product No. 47768) and vitamin D3 (product No. 47763) were obtained from Sigma-Aldrich (St. Louis, MO, USA).

### 2.2. Selection and Collection of Freshwater and Marine Fish 

Five species of the most commonly consumed freshwater fish and four species of the most commonly consumed marine fish were selected based on data from the fisheries statistics of Thailand [11] and data of commonly consumed foods from the National Food Consumption Survey [12]. Table 1 includes the names of the selected fish and their characteristics. Sample collection was conducted using simple random sampling (SRS) and followed the guidelines for food composition database development [13]. Three single composite samples (*n* = 3), for each of the five species of freshwater fish, were purchased from three representative markets: (1) Klong-toey market, Klong-toey district, Bangkok (a representative of the Bangkok area), (2) Ying-cha-roen market, Bang Khen district, Bangkok (a representative of northern and/or north-eastern parts of Thailand), and (3) Nakhon Pathom market, Nakhon Pathom province (a representative of southern and/or western part of Thailand). Another three single composite samples (*n* = 3) for each of four species of marine fish were purchased from three markets under the Fish Marketing Organization (FMO) located in Bangkok, Samutsakorn, and Chonburi provinces (fish samples came from the Gulf of Thailand). All of the purchased fish were without eggs, except for the Common silver barb, for which the eggs were removed from the raw and cooked fish before nutrient analysis. The width and length of four individual fish of the same variety from three markets (*n* = 12) were measured by a metal ruler. For the weight, a top pan precision balance was used. The somatometry of the selected commonly consumed freshwater and marine fish are included in Table 1 for information. 

### 2.3. Preparation and Cooking Processes 

The collected fish samples for each of the individual species from each source were divided into four groups of equal size and number. Sample preparation for each species as raw and cooked fish before and after heat treatments differed depending on the characteristics of each fish and the most common way of cooking. Before sample preparation, all fresh fish were rinsed thoroughly under running water to remove adhering blood and mucus. For raw fish, they were prepared using household practices (i.e., eviscerating, descaling (except for Walking catfish and Short-bodied mackerel, which have no scales), without removing the skin) and washed again twice with deionized water. Edible portions (including flesh from the head) with skin were collected, and inedible portions were discarded. 

For cooking the fish, three heating processes were applied: boiling, frying, and grilling. Table 2 shows the cooking processes used for the three individual sets (*n* = 3) of each fish. The internal temperature of each cooking method was measured using a digital thermometer (Oakton, TEMP10J, NJ, USA). The fish were cooked without adding any other ingredients or seasonings. 

For boiled fish, the prepared raw whole fish (with the head on) was placed in a pot with two liters of boiling water until cooked. The cooking time for each fish was recorded. Based on household practices and recipes, the Common silver barb required a longer cooking time of 2 h. This fish must then be left overnight at room temperature and simmered for another 2 h until the bones soften. After cooking by boiling, the cooked fish was drained for 30 s to remove excess water before preparing the edible portion, which included the flesh with the soft bone.

For fried fish, a liter of palm oil was used as the cooking medium. Large whole fish, such as Striped snakehead and Walking catfish, were sliced and cut into suitably sized pieces before frying. Since the Common silver barb has numerous small bones, it was cut into strips of about 0.5 cm in width before frying. After frying, fish with crispy, small bones can be consumed. Cutting into strips for the smaller-sized Short-bodied mackerel was not necessary. The oil was heated before deep-frying. One fish at a time for all varieties was fried, except that of the Short-bodied mackerel, where three whole fish were fried with 50 milliliters of the palm oil. The oil was changed after frying each fish. All fish were fried until completely cooked and crispy. 

For traditional Thai grilling, washed whole fish (without descaling) were grilled on a stainless steel grate above a charcoal stove. Before grilling, the charcoal was burned down until ashen. The heat from the charcoal was controlled by distributing the ash before and during grilling. Grilling continued until the fish was well-cooked. For grilling fish with scales, the scales and skin from the cooked fish were removed before being prepared as edible portions. 

After cooking, only the edible portions of the cooked fish were homogenized in a food blender (Wongdec, WTI-1684A, Nonthaburi, Thailand) and divided into two parts. The first part was kept in a clean screwed-cap bottle for moisture analysis. The second part was freeze-dried using a freeze dryer (Heto Powerdry PL 9000, Corston, UK). After drying, the freeze-dried fish was re-homogenized with a food blender and packed in about 50 g portions in aluminum foil bags and kept at −20 °C for vitamin D analysis.

### 2.4. Determination of Vitamin D Content 

AOAC Official Method number 995.05 [14] was used to determine vitamin D content in commonly consumed freshwater and marine fish. The method was previously described in detail by Tirakomonpong et al., 2019 [10]. In brief, the lyophilized sample (1.0 ± 0.3 g) was saponified with ethanolic potassium hydroxide in a shaking water bath (Brunswick Scientific G76 Gyrotory Shaker Water Bath, New Hartford, CT, USA), and then vitamin D was extracted with hexane. The extracted solution was evaporated using a rotary evaporator (BUCHI R-114 Rotary Vap System, Flawil, Switzerland) and purified by solid-phase extraction using a silica column (Cleanert^®^ SPE, Walnut, CA, USA, 500 mg/3 mL). The eluted fraction was injected (250 µL) and separated by High Pressure Liquid Chromatography (Agilent 1100 Series HPLC system, Palo Alto, CA, USA). HPLC key components consisted of an isocratic pump (G1310A), diode array detector (DAD-MWD/G1365B), a 250 µL injector loop, and reversed-phase C18 column (4.6 × 250 mm, 5 µm). For quantitation of vitamin D3, vitamin D2 was used as an internal standard. One milliliter of vitamin D2 (2.9 µg/mL) was added to each concentration of vitamin D3 standards (0.1–0.6 µg/mL) and to each sample before saponification. Vitamin D3 in the unknown samples was quantified from the standard curve and reported as µg per 100 g edible portion (EP) of a raw or cooked sample. 

### 2.5. Method Validation and Quality Control System for Vitamin D Analysis 

Method validation of vitamin D analysis in fish samples that includes the limit of detection (LOD), limit of quantitation (LOQ), accuracy, precision, intermediate precision, internal quality control system, and the interlaboratory performance study was previously conducted and reported by Tirakomonpong et al. (2019) [10]. Fortified vitamin D3 milk powder, with assigned values of 5.2 ± 0.5 μg/100 g (%RSD 9.6%), was used as an in-house quality control (QC) sample. Each fish sample was analyzed in duplicate along with the QC sample. The vitamin D values of the QC samples obtained from each batch of analysis must be within the mean ± 2 standard deviations (SD) of the assigned values.

### 2.6. True Retention of Vitamin D in Cooked Fish

To investigate the effect of cooking on vitamin D in the cooked fish, the weight of each fish sample (3 significant digits)—before and after cooking—was recorded. The true retention [15] of vitamin D in cooked fish by different cooking methods was calculated as follows.
$$\%\mathrm{True}\;\mathrm{retention}=\frac{µg\;\mathrm{vitamin}\;D\;\mathrm{per}\;100\;g\;\mathrm{of}\;\mathrm{cooked}\;\mathrm{fish}\;\times \;\mathrm{weight}\;\mathrm{of}\;\mathrm{cooked}\;\mathrm{fish}}{µg\;\mathrm{vitamin}\;D\;\mathrm{per}\;100\;g\;\mathrm{of}\;\mathrm{raw}\;\mathrm{fish}\;\times \;\mathrm{weight}\;\mathrm{of}\;\mathrm{raw}\;\mathrm{fish}}\;\times \;100$$

### 2.7. Statistical Analysis

Vitamin D values in raw and cooked freshwater and marine fish were reported as the mean ± standard deviation. The statistical significance of vitamin D contents in different types of fish and their true retention after cooking were assessed (types of fish and cooking methods), followed by Tukey’s Honestly Significant Difference to test multiple pairwise comparisons. Statistical analysis was performed using IBM^®^ (Hampshire, UK) SPSS Statistics for Window, Version 19.0. 

## 3. Results 

### 3.1. Edible Portion and Yield Factor

The edible portion of food is the proportion of edible matter in raw food as collected or as purchased, or the edible portion in cooked food expressed on a wet weight basis. The percentage of the edible portion of raw and cooked freshwater and marine fish, presented on a wet weight basis, is shown in Table 3 and Table 4 respectively. 

Yield factor is the percentage of weight change in foods or recipes due to cooking. For this study, boiled and roasted freshwater and marine fish retained a high yield factor ranging from approximately 0.8 to 0.9 and 0.6 to 0.8 (Table 3 and Table 4), respectively. 

### 3.2. Moisture Content 

Vitamin D content in fish is presented on a fresh basis. Initial moisture contents of raw and cooked fish using different methods can affect vitamin D content per 100 g. The moisture contents of raw and cooked freshwater and marine fish are shown in Table 3 and Table 4 respectively. 

### 3.3. Vitamin D Content 

Vitamin D3 was the only detectable form of vitamin D in the studied fish. Table 3 shows the vitamin D content of raw and cooked freshwater fish. Three types of raw freshwater fish, namely Common silver barb, Red Nile tilapia, and Nile tilapia, contained extraordinarily high levels of vitamin D (48.5, 31.0, and 19.8 µg per 100 g EP, respectively). Very low levels of vitamin D were found in the raw marine fish (2.94 to 4.69 µg per 100 g EP) (Table 4). 

After testing the data of vitamin D in fish by two-way ANOVA with interaction followed by Tukey’s HSD post hoc test, it revealed no significant differences in the combined effects of the species of fish and the cooking methods (*p* > 0.05) (Figure 1A). These findings suggest that the effect of cooking method on vitamin D content was not significantly different among different species of fish. Significant differences were found in vitamin D content among different kinds of fish (*p* < 0.001) (Table 5). Common silver barb and Red Nile tilapia showed significantly higher vitamin D levels than other fish (*p* ≤ 0.05). In addition, there was no significant effect of cooking method on vitamin D content (*p* > 0.05) (Table 5, estimated marginal means ranged from 15.2 to 17.5 µg per 100 g EP).

### 3.4. Effect of Different Cooking Methods on Vitamin D Retention

The percentage of true retention (TR) of vitamin D in cooked fish is presented in Table 3. Most fish cooked by boiling retained vitamin D, ranging from 77% in Walking catfish to 100% in Red Nile tilapia and Grey mullet. Due to prolonged boiling time (240 min) to soften the bone of the Common silver barb, a lower %TR was found (66 %TR). Cooking by frying retained vitamin D in most fish (75–100 %TR), except for Common silver barb (64 %TR), Walking catfish (51 %TR), and Striped snakehead (22 %TR). Most grilled fish, not covered with foil, showed true retention in the range of 78 to 100%TR. However, Giant sea perch and Short-bodied mackerel showed lower true retention of vitamin D (49 and 47 %TR) compared to the other species. 

Results from two-way ANOVA with interaction followed by Tukey’s HSD post hoc test, significant differences of the combined effects of different kinds of fish and cooking methods on %TR of vitamin D were found (*p* ≤ 0.05). The significant main effect of different kinds of fish on %TR was found (*p* ≤ 0.05), as shown in Table 6. Figure 1B shows the combined effects of kinds of fish and cooking methods and demonstrates that cooking methods affect the different magnitudes of %TR in different kinds of fish. Only fried Striped snakehead (lowest %TR) showed significantly lower %TR than that of Giant sea perch, Grey Mullet, Black-banded trevally, and Short-bodied mackerel (*p* ≤ 0.05). Boiled and grilled fish of all varieties showed no significant differences of %TR (*p* > 0.05) (Table 6). Another main effect of cooking methods, it was found that all cooking methods have no significant differences of %TR (*p* > 0.05) (Table 5). 

## 4. Discussion

### 4.1. Edible Portion and Yield Factor

The edible portion of the raw freshwater fish (46–51%) was less than that of the raw marine fish (52–86%). Different edible portions depend on the variety of fish species. The lower amount of bone present in marine fish may explain its higher edible portion. A similar finding of the edible portion (i.e., 50–58% freshwater fish and 61–73% marine fish) was noted by Bilodeau et al. (2011) [16]. Cooked freshwater and most marine fish contained a higher percentage of edible portion than raw fish. The cooking process can facilitate the separation of the edible part of the fish from the bone. 

Cooking yields obtained from this study compare well with those presented by Bognár (2002) [17]. Boiled Common silver barb showed a lower yield factor than other fish due to loss of flesh by a long cooking process (4 h). Different yield factors are affected by treatment conditions and muscle hardness in fish. Moisture loss during frying provided a low percent yield for both freshwater and marine fish. Percentage of edible portion and yield factor are fundamental factors for accurate recipe calculation in inputting nutritive values of cooked foods (dishes). 

### 4.2. Moisture Content

Raw freshwater and marine fish moisture ranged from 68 to 76 and 72 to 77 g per 100 g EP, respectively. These results agree well with a previous study [16] of raw Walking catfish, Nile tilapia, Common silver barb, Sea bass, Black-banded trevally, and Short-bodied mackerel (65, 78, 73, 79, 71, and 73 g per 100 g EP, respectively). In addition, these findings agree with those reported by the Food and Agriculture Organization of the United Nations (2016) [18] for raw freshwater, diadromous (Salmons and Rainbow trout), and marine fish (ranging from 66 to 81 g per 100 g EP).

Both boiled and grilled freshwater and marine fish showed slightly lower moisture contents (1.4–6.6% and 0.0–13.2%, respectively) than raw fish. The cooking conditions of roasted fish with scales could protect against water loss during cooking. Frying using a higher heat temperature was the major cause of water loss (11.8–45.9%) in fried fish, resulting in lower moisture than in other samples.

### 4.3. Vitamin D Content 

Freshwater fish with a high vitamin D content include varieties living in the limnetic zone populated by phytoplankton and zooplankton (i.e., Common silver barb, Red Nile tilapia, and Nile tilapia). The other two studied species of freshwater fish (i.e., Striped snakehead and Walking catfish) that contained lower vitamin D live in the profundal area or in the mud of a pond located beyond the range of sunlight penetration. Vitamin D contents in Common silver barb and Red Nile tilapia were significantly (*p* ≤ 0.05) higher than those of other freshwater and marine fish. Rao and Raghuramulu (1999) [19] found high levels of provitamin D3 in phytoplankton and zooplankton (2358 and 4624 µg per 100 g dry weight, respectively) and vitamin D3 (80 and 272 µg per 100 g dry weight, respectively). Consequently, the high level of vitamin D found in fish could be derived from plankton photosynthesizing vitamin D [19]. Another study [20] reported a lower vitamin D3 content in plankton from various lakes and the Baltic Sea (1.5–6.9 and 10.5 µg per 100 g, respectively). From the reported findings, plankton may not be the only key factor affecting vitamin D content in fish. Visible light from the sun that induces cholecalciferol production in the skin of fish living in the limnetic zone may also play an important role.

The lack of a database and limited data on vitamin D levels in commonly consumed fish in Thailand, and especially the effects of cooking, limits species comparisons. Reports from other countries, however, can provide some insights into this issue. In a previous study in Canada, Bilodeau et al. (2011) [16] reported vitamin D in three different fish, namely Mahi-mahi, Canned pink salmon, and Tilapia, at average levels of 1.11 (0.24–2.24), 22.3 (12.7–43.5), and 45.3 (17.9–75.3) µg per 100 g EP, respectively. For this study, Nile tilapia (with skin) contained a vitamin D level (19.8 µg per 100 g EP) lower than that reported in a previous Canadian study. Further, the Food and Agriculture Organization of the United Nations (2016) [18] reported that vitamin D levels in Nile tilapia from China, Bangladesh, and Africa ranged from 19.95 to 20.15 µg per 100 g EP. Frequently consumed fish and their vitamin D levels in the Japanese diet are salmon (10 to 32 µg per 100 g), flat fish (3 to 18 µg per 100 g), sea bream (1 to 8 µg per 100 g), and mackerel (1 µg per 100 g) [21]. Vitamin D3 levels in Atlantic salmon fillets without skin from Norway, Iceland, Ireland, Chile, Northeast Atlantic, and Northwest Atlantic ranged from 4.14 to 14.43 µg per 100 g EP [18]. In addition, a wide range of vitamin D levels (1.8 to 30 µg per 100 g) was found in the skin of different types of fish, namely, Trevally, Atlantic salmon, Yellowfin tuna, Bream, Blackfish, Whiting, and Rainbow trout [22]. In general, vitamin levels can vary in different parts of the same tissues, as well as among animals collected at different times and locations. Indeed, geographic availability, seasonality, and physiological state/maturity are known factors that affect variability in nutrient composition, particularly for vitamins [23]. A previous study assumed that fish, especially oily fish, such as salmon, mackerel, and bluefish, are excellent sources of vitamin D3 [8]. However, this study found that oily fish, such as Striped snakehead, Walking catfish, and Black-banded trevally (fat content ranged from 5.97, 13.06, and 5.87 g per 100 g EP, respectively), contained low levels of vitamin D3. Mattila et al. (1997) [20] reported no consistent relationship between the weight, sex, age, and fat content of fish and cholecalciferol content. They concluded that diet is a likely factor that causes variations in the cholecalciferol contents of the fish.

Cooked freshwater fish contained higher vitamin D content than cooked marine fish, ranging from 1.6 to 30.3 µg per 100 g EP and 0.4 to 7.7 µg per 100 g EP, respectively. Although cooking methods showed no significant differences in vitamin D content (*p* > 0.05, two-way ANOVA), significant differences in the vitamin were found among different kinds of fish (*p* < 0.001). This indicated that the raw fish with higher vitamin D content (such as Common silver barb, Red Nile tilapia, and Nile tilapia) provided high vitamin D after cooking by boiling, frying, and grilling. A previous study also reported vitamin D3 contents in boiled and grilled Nile tilapia from China, Bangladesh, and Africa, ranging from 20.64 to 22.96 µg per 100 g EP, respectively [18]. The wide range of vitamin D in cooked fish could be due to differences in fish species, season, geographic availability, physiological state, and cooking methods [23]. 

### 4.4. Effect of Different Cooking Methods on Vitamin D Retention

High true retention of vitamin D (77–100%) was shown to be insignificantly different (*p* > 0.05) among various types of boiled fish. This finding indicates that cooking by boiling retained significant levels of vitamin D in all types of fish.

Applying different cooking methods to different species of fish caused a lower %TR of vitamin D at different levels in cooked fish. Vitamin D in fried freshwater fish, especially fried Striped snakehead (22 %TR), Walking catfish (51 %TR), and Common silver barb (64 %TR), retained vitamin D lower than that of the fried marine fish (92–97% TR). The true retention of fried marine fish showed no significant difference from that of raw fish. Ložnjak and Jakobsen (2018) [24] investigated the retention of vitamin D in rainbow trout using different cooking methods—boiling at different pH, steaming, microwave cooking, pan-frying, and oven baking. The true retention of vitamin D was 85–114%, which was not significantly different (*p* > 0.05) from the raw fish. Loss of vitamin D during frying could be associated partly with the destruction of vitamin D by high temperature (110–150 °C) and partly with leaching into the cooking oil during deep frying. 

Although most grilled fish was not covered with foil during the process of grilling, vitamin D was retained at 78 to 100% of the raw fish. However, Giant sea perch and Short-bodied mackerel showed significantly lower (*p* ≤ 0.05) true retention of vitamin D (49 and 47 %TR) compared to those of other fish. The higher loss of vitamin D was possibly due to grilling the fish without scales, long grilling time for Giant sea perch (45–55 min), and the small size of the Short-bodied mackerel (21.4 ± 1.4 cm. in length and 5.2 ± 0.2 cm in width). Mattila et al. (1999) [25] reported that rainbow trout, covered with foil and treated in an oven at 172–200 °C for 20 min, retained vitamin D in the range of 85 to 114%. 

## 5. Conclusions

This study aimed to determine vitamin D in selected commonly consumed fresh water and marine fish in Thailand and its effect on cooking. Freshwater fish living in the limnetic zone (Common silver barb, Red Nile tilapia, and Nile tilapia) contained significantly higher levels of vitamin D compared to those living in the profundal area or the mud of a pond (Striped snakehead and Walking catfish). However, the effects of seasonality, sun exposure of the fish, and geographic variability (diet, plankton exposure) on vitamin D content in fish have not yet been assessed in this study. Vitamin D content was lower in raw and cooked marine fish than in freshwater fish. Vitamin D retention in boiled fish was higher than in fried and grilled fish. The true retention of vitamin D in all varieties of boiled fish was not significantly different from the level found in raw fish, except for the Common silver barb that could be due to prolonged boiling time (4 h) to soften the bone. Vitamin D retention in fried fish was not significantly different from the raw, except that of the fried Striped snakehead, Walking catfish, and Common silver barb. These fish were sliced and cut into small pieces before frying, which could increase surface area and cause a higher loss of vitamin D during deep frying and leaching into the cooking oil. Common silver barb, Red Nile tilapia, and Nile tilapia contained high vitamin D content among the studied fish, and no significant loss after cooking by boiling, frying, and grilling. In contrast, raw Striped snakehead and Walking catfish, which contained low vitamin D contents, appeared to have a high loss after frying. Three freshwater fish—Common silver barb, Red Nile tilapia, and Nile tilapia—cooked by boiling, frying, and grilling are recommended for consumption as excellent sources of vitamin D. 

## Figures and Tables

**Figure 1 foods-11-00819-f001:**
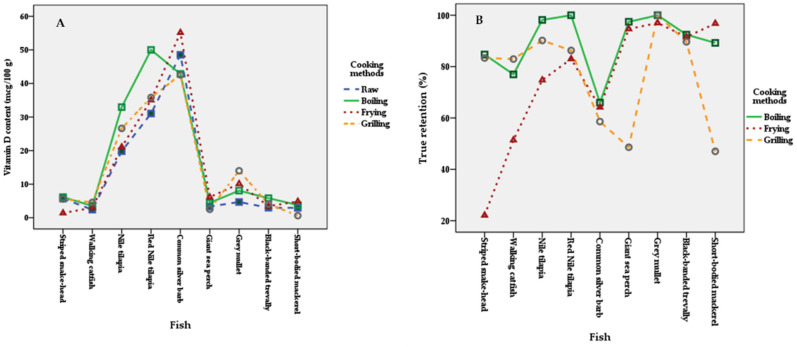
The combined effects of different kinds of fish and cooking methods on vitamin D content (**A**) and on the percentage of true retention (**B**).

**Table 1 foods-11-00819-t001:** Names and characteristics of selected commonly consumed freshwater and marine fish.

English Name	Fish with Scales	Local Name	Scientific Name	Characteristics (Mean ± SD of Four Individual Fish of Each Variety from Three Markets (*n* = 12))		
				Length (cm)	Width (cm)	Weight (g)
Freshwater fish:						
Striped snakehead	Yes	Pla-chon	*Channa striatus*	39.1 ± 3.4	6.2 ± 0.7	649 ± 119
Walking catfish	No	Pla-duk	*Clarias macrocephalus*	38.7 ± 3.4	5.9 ± 0.5	477 ± 102
Nile tilapia	Yes	Pla-nin	*Oreochromis niloticus*	29.7 ± 2.4	11.3 ± 0.5	571 ± 84
Red Nile tilapia	Yes	Pla-tub-tim	*Oreochromis niloticus-mossambicus*	31.6 ± 1.2	12.3 ± 0.7	749 ± 86
Common silver barb	Yes	Pla-ta-pian	*Puntius gonionotus*	25.6 ± 5.7	7.9 ± 2.4	292 ± 75
Marine fish:						
Giant sea perch	Yes	Pla-kha-pong-khaw	*Lates calcarifer*	37.5 ± 1.1	10.8 ± 0.4	775 ± 87
Grey mullet	Yes	Pla-kha-boak	*Mugil cephalus*	36.7 ± 2.7	7.3 ± 0.8	522 ± 101
Black-banded trevally	Yes	Pla-sum-lee	*Seriolina nigrofasciata*	33.4 ± 3.8	8.6 ± 1.0	532 ± 141
Short-bodied mackerel	No	Pla-tu	*Rastrelliger brachysoma*	21.4 ± 1.4	5.2 ± 0.2	113 ± 23

**Table 2 foods-11-00819-t002:** Cooking time for three individual sets (*n* = 3) of each fish using different cooking methods.

Fish Name	Cooking Time (min) for Different Cooking Methods		
	Boiling (100 °C)	Frying (110–150 °C)	Grilling (230–250 °C)
Striped snakehead	8–10	8–10	40–45
Walking catfish	8–9	5–7	40–45 *
Nile tilapia	5–7	7–8	30–45
Red Nile tilapia	7–8	7–8	30–45
Common silver barb	240	6–8	40
Giant sea perch	3–6	8–10	45–55
Grey mullet	5–6	7–8	35–45
Black-banded trevally	7–8	7–8	30–40
Short-bodied mackerel	4–5	5–7	6–10 *

* Edible portion includes skin.

**Table 3 foods-11-00819-t003:** Percentage of edible portion, yield factor, moisture, vitamin D, and true retention of 3 individual sets from each type of freshwater fish, data expressed as mean ± SD (*n* = 3).

Species of Fish	Type of Sample	Edible Portion (%)	Yield Factor	Moisture (g/100 g)	Vitamin D (µg/100 g EP)	True Retention of Vitamin D (%)
Striped snakehead	Raw (with skin)	50 ± 3	-	74 ± 0.4	5.7 ± 2.6	-
	Boiled (with skin)	56 ± 4	0.90 ± 0.04	72 ± 1.3	6.2 ± 2.8	85 ± 27
	Fried (with skin)	41 ± 3	0.65 ± 0.02	57 ± 1.1	1.4 ± 0.2	22 ± 11
	Grilled (skinless)	51 ± 1	0.88 ± 0.03	69 ± 4.6	5.6 ± 2.6	83 ± 29
Walking catfish	Raw (with skin)	51 ± 6	-	68 ± 0.6	2.4 ± 1.4	-
	Boiled (with skin)	58 ± 2	0.92 ± 0.02	64 ± 2.3	3.5 ± 3.1	77 ± 40
	Fried (with skin)	41 ± 4	0.70 ± 0.04	54 ± 7.8	3.0 ± 3.0	51 ± 43
	Grilled (with skin)	50 ± 5	0.81 ± 0.07	67 ± 7.4	4.6 ± 2.6	83 ± 30
Nile tilapia	Raw (with skin)	46 ± 6	-	76 ± 1.8	19.8 ± 3.5	-
	Boiled (with skin)	53 ± 4	0.90 ± 0.04	73 ± 1.1	33.0 ± 11.1	98 ± 3
	Fried (with skin)	39 ± 3	0.70 ± 0.00	57 ± 0.6	21.1 ± 6.3	75 ± 16
	Grilled (skinless)	43 ± 3	0.81 ± 0.04	72 ± 1.9	26.6 ± 7.2	90 ± 17
Red Nile tilapia	Raw (with skin)	50 ± 2	-	73 ± 0.4	31.0 ± 7.7	-
	Boiled (with skin)	60 ± 7	0.91 ± 0.02	70 ± 2.3	50.0 ± 12.2	100 ± 0
	Fried (with skin)	44 ± 5	0.71 ± 0.03	59 ± 3.0	35.1 ± 7.1	83 ± 29
	Grilled (skinless)	46 ± 1	0.84 ± 0.01	70 ± 0.5	35.8 ± 7.2	86 ± 24
Common silver barb	Raw (with skin)	50 ± 7	-	74 ± 3.2	48.5 ± 26.5	-
	Boiled (with skin)	56 ± 3	0.77 ± 0.08	71 ± 1.1	42.8 ± 12.8	66 ± 31
	Fried (with skin)	41 ± 9	0.55 ± 0.03	40 ± 2.7	55.2 ± 36.9	64 ± 31
	Grilled (skinless)	48 ± 5	0.84 ± 0.06	72 ± 1.3	42.6 ± 33.6	59 ± 38

**Table 4 foods-11-00819-t004:** Percentage of edible portion, yield factor, moisture, vitamin D, and true retention of 3 individual sets from each type of marine fish, data expressed as mean ± SD (*n* = 3).

Species of Fish	Type of Sample	Edible Portion (%)	Yield Factor	Moisture (g/100 g)	Vitamin D (µg/100 g EP)	True Retention of Vitamin D (%)
Giant sea perch	Raw (with skin)	54 ± 3	-	74 ± 3.0	3.3 ± 2.8	-
	Boiled (with skin)	65 ± 3	0.77 ± 0.22	73 ± 1.8	4.5 ± 2.8	97 ± 4
	Fried (with skin)	43 ± 3	0.66 ± 0.04	54 ± 2.0	6.2 ± 4.7	95 ± 9
	Grilled (skinless)	52 ± 11	0.83 ± 0.01	74 ± 0.7	2.5 ± 2.5	49 ± 8
Grey mullet	Raw (with skin)	53 ± 1	-	77 ± 0.6	4.7 ± 0.8	-
	Boiled (with skin)	57 ± 2	0.62 ± 0.09	74 ± 3.8	8.1 ± 1.3	100 ± 0
	Fried (with skin)	41 ± 5	0.81 ± 0.02	59 ± 8.0	10.1 ± 3.6	97 ± 5
	Grilled (skinless)	42 ± 4	0.84 ± 0.01	72 ± 2.6	14.0 ± 5.2	100 ± 0
Black-banded trevally	Raw (with skin)	58 ± 6	-	72 ± 1.8	3.0 ± 1.3	-
	Boiled (with skin)	54 ± 5	0.81 ± 0.06	69 ± 2.5	5.8 ± 2.8	92 ± 13
	Fried (with skin)	47 ± 7	0.71 ± 0.04	57 ± 1.1	3.4 ± 2.7	92 ± 12
	Grilled (skinless)	46 ± 7	0.76 ± 0.08	71 ± 2.4	4.1 ± 0.5	90 ± 18
Short-bodied mackerel	Raw (with skin)	52 ± 5	-	76 ± 1.8	2.9 ± 2.1	-
	Boiled (with skin)	47 ± 3	0.83 ± 0.06	71 ± 2.4	3.8 ± 2.9	89 ± 19
	Fried (with skin)	41 ± 3	0.77 ± 0.10	67 ± 4.9	4.8 ± 3.5	97 ± 6
	Grilled (with skin)	41 ± 1	0.74 ± 0.04	66 ± 5.1	0.6 ± 0.1	47 ± 13

**Table 5 foods-11-00819-t005:** Estimated marginal means of vitamin D content and percentage of vitamin D true retention by the main effects of different species of fish and cooking methods (calculated from two-way ANOVA) (*n* = 3).

Variables	Estimated Marginal Means ± Standard Error	
	Vitamin D (µg/100 g EP)	True Retention (%)
Different Species of Fish:		
Striped snake-head	4.7 ± 3.1 ^c^	63.4 ± 7.8
Walking catfish	3.4 ± 3.1 ^c^	70.4 ± 7.8
Nile tilapia	25.1 ± 3.1 ^b^	87.7 ± 7.8
Red Nile tilapia	38.0 ± 3.1 ^a,b^	89.8 ± 7.8
Common silver barb	47.3 ± 3.1 ^a^	62.9 ± 7.8
Giant sea perch	4.1 ± 3.1 ^c^	80.2 ± 7.8
Grey Mullet	9.2 ± 3.1 ^c^	99.0 ± 7.8
Black-banded trevally	4.1 ± 3.1 ^c^	91.2 ± 7.8
Short-bodied mackerel	3.1 ± 3.1 ^c^	77.7 ± 7.8
Cooking methods in different species of fish:		
Raw	13.5 ± 2.0 ^a^	-
Boiling	17.5 ± 2.0 ^a^	89.4 ± 4.5
Frying	15.6 ± 2.0 ^a^	75.0 ± 4.5
Grilling	15.2 ± 2.0 ^a^	76.3 ± 4.5

Values with different superscript letters of species of fish or cooking methods in the same column were significantly different for a given variable (*p* < 0.05 two-way ANOVA followed by Tukey’s HSD post hoc multiple comparisons).

**Table 6 foods-11-00819-t006:** Estimated marginal means of interaction effect of species of fish and cooking methods on true retention of vitamin D (calculated from two-way ANOVA) (*n* = 3).

Species of Fish	True Retention of Vitamin D (Mean ± Standard Error (SE)		
	Boiled	Fried	Grilled
Striped snake-head	84.7 ± 11.8 ^g^	22.1 ± 12.7 ^a,b,c,d^	83.4 ± 15.5 ^e^
Walking catfish	77.0 ± 11.8 ^h^	51.4 ± 12.7 ^h^	82.9 ± 12.7 ^f^
Nile tilapia	98.2 ± 11.8 ^c^	74.8 ± 12.7 ^f^	90.2 ± 12.7 ^b^
Red Nile tilapia	100.0 ± 11.8 ^b^	83.0 ± 12.7 ^e^	86.3 ± 12.7 ^d^
Common silver barb	66.0 ± 11.8 ^i^	64.2 ± 12.7 ^g^	58.6 ± 12.7 ^g^
Giant sea perch	97.4 ± 11.8 ^d^	94.6 ± 12.7 ^c^	48.6 ± 12.7 ^h^
Grey Mullet	100.0 ± 11.8 ^a^	97.0 ± 12.7 ^a^	100.0 ± 12.7 ^a^
Black-banded trevally	92.4 ± 11.8 ^e^	91.6 ± 12.7 ^d^	89.6 ± 12.7 ^c^
Short-bodied mackerel	89.2 ± 11.8 ^f^	96.8 ± 12.7 ^b^	47.0 ± 12.7 ^i^

Estimated marginal means values with the same superscript letters in the same column were significantly different for a given variable (*p* < 0.05 two-way ANOVA followed by Tukey’s HSD post hoc multiple comparisons).

## Data Availability

Data are contained within this article.

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
