# Peer review of "Effects of Different Cooking Methods on the Vitamin D Content of Commonly Consumed Fish in Thailand"

_foods, 2022, doi:10.3390/foods11060819_

Round 1

Reviewer 1 Report

Introduction

the introduction is sufficiently informative, has a good structure and underlines the novelty meaning of the study.

Materials and Methods

Materials and methods are analytically, but not exhaustively, presented in a way that all processes can be checked and reproduced. All analytic methods are sound. The main method of analysis, that of vitamin D determination is analytically described in the citation of Tirakomonpog et al. (2019) and can be easily obtained.

Some issues, however, need further justification.  a) Not clear how many individuals were included in every sampling. The n=3 is supposingly the number of markets where from fish were obtained. Authors should clarify number of individuals measured for somatometry and analyzed in every cooking method, number of samplings is not an adequate element. b) The Statistical method that the authors used, that of t-test is to compare two groups of samples, not for multiple comparisons among different groups. A minimum of ANOVA analysis would allow simple comparisons. A far more sophisticated statistical method is required herein in my opinion. A two-way ANOVA with "cooking method" as one factor and "species" as a second factor (the two independent variables). The currently used statistic technique severely weakens the quality of this manuscript. Thus I strongly suggest

Results and discussion.

The results and discussion would definitely posititively affected if the correct statistical analysis takes place. The effect of cooking and inter-species comparison as well as the combined effects of cooking and species would give a much more depth to the conclusions driven from this study.

Therefore, in overall, although I can see the scientific importance of the study, the sound study design and the amount of lab work it contains, the weak statistics, and their subsequent effect on results and discussion, cannot allow publication of this manuscript in its current form.

Author Response

Dear Editor and Reviewers of Foods

According to the comments and suggestions from reviewers, all authors appreciated very much for valuable comments and suggestions.  We revised all addressed issues from all reviewers. Now, may I submit this revised manuscript for further consideration.

I appreciate very much your kind consideration of this manuscript for publication.

Yours sincerely

Kunchit Judprasong

Reviewer 2 Report

In this manuscript, the authors studied the vitamin D content in commonly consumed five species of fresh water fish and four species of marine fish in Thailand and compared the effect of different cooking methods (boiling vs. frying vs. grilling) on vitamin D retention. I evaluated carefully the study by Sridonpai et al. There are some comments regarding the manuscript.

1) The authors wrote the results and discussion part together making the manuscript complicated to understand. The authors had better separate the parts of the manuscript.

2) Please report the the p values of the analysis of Table 4,  both among the groups of the fish and in the groups of the cooking methods of the fish.  

3) Limitations such as the season, sun exposure of the fish, geographic variability (diet, plankton exposure) should be added to the manuscript.

4) There is a general lack of discussion of the results from the previous studies. The data presented in the manuscript should be discussed together with the potential explanations for the differences in the findings.

5) It needs language editing.

Author Response

(The authors gave the same response as above.)

Round 2

Reviewer 1 Report

The manuscript has been slightly improved. However, I still see some major drawbacks. Values are often duplicated in tables and text. This should be fixed.

I still see no result & discussion on the effect of each factor as well as on the combined effect of factors. Other than mentioning that they did two-way ANOVA statistics in the Materials and Methods, this does not reflect to the results and discussion.

(suggestion: table with a) "species" and b) "cooking method" as independent variables and c) "species*cooking combined effect" and percentage of total variation explained by those factors) - relative discussion has to be made.

Author Response

Manuscript ID: foods-1593813
Type of manuscript: Article
Title: Effects of different cooking methods on the vitamin D content of commonly consumed fish in Thailand
Authors: Piyanut Sridonpai, Kunchit Judprasong *, Nichaphan Tirakomonpong,
Preecha Saetang, Prapasri Puwastien, Nipa Rojroongwasinkul, Boonsong
Ongphiphadhanakul

Dear Editor and Reviewers of  “Foods”

All authors appreciated very much for valuable comments and suggestions from the reviewers.  For this revision, we revised the manuscript by adding the output from two-way ANOVA analysis in the results and discussion.  Please see the revised manuscript in the journal system for further consideration.

I appreciate very much your kind consideration of this manuscript for publication.

Yours sincerely

Kunchit Judprasong

Kunchit Judprasong, Ph.D.

Associate Professor

Institute of Nutrition, Mahidol University

Putthamonthon 4, Salaya, Nakhon Pathom 73170, Thailand

Tel.: +66 2800 2380; Fax: +66 2441 9344

E-mail address: Kunchit.jud@mahidol.ac.th

Response to Reviewers

Comments and Suggestions for Authors

Response to Comments and Suggestions

The manuscript has been slightly improved. However, I still see some major drawbacks. Values are often duplicated in tables and text. This should be fixed.

The mean values of all parameters are used in the Results section.

I still see no result & discussion on the effect of each factor as well as on the combined effect of factors. Other than mentioning that they did two-way ANOVA statistics in the Materials and Methods, this does not reflect to the results and discussion.

- Results and discussion based on two-way analysis of variance (ANOVA) (types of fish and cooking methods), followed by Tukey's Honestly Significant Difference are added in the Results and discussion of the revised manuscript.

- Figures of the combined effects between different kinds of fish and cooking methods were prepared and added to the revised manuscript  in order to clearly demonstrate  the combined effects.

(suggestion: table with a) "species" and b) "cooking method" as independent variables and c) "species*cooking combined effect" and percentage of total variation explained by those factors) - relative discussion has to be made.

Symbols in Table 4:  small letters and Capital letters are used to clearly separate between species of fish (small letters) and cooking methods (capital letters).  Small letters are used to indicate significant differences among different kinds of fish whereas capital letters are used to indicate significant differences among cooking methods. 

So, the authors decided to keep the same symbol system in the manuscript.
